# PlantMetSuite: A User-Friendly Web-Based Tool for Metabolomics Analysis and Visualisation

**DOI:** 10.3390/plants12152880

**Published:** 2023-08-06

**Authors:** Yu Liu, Hao-Zhuo Liu, Ding-Kang Chen, Hong-Yun Zeng, Yi-Li Chen, Nan Yao

**Affiliations:** State Key Laboratory of Biocontrol, Guangdong Provincial Key Laboratory of Plant Resource, School of Life Sciences, Sun Yat-sen University, Guangzhou 510275, China; liuy233@mail2.sysu.edu.cn (Y.L.); liuhzh35@mail2.sysu.edu.cn (H.-Z.L.); cdkon@outlook.com (D.-K.C.); zenghy7@mail2.sysu.edu.cn (H.-Y.Z.); chenyli25@mail2.sysu.edu.cn (Y.-L.C.)

**Keywords:** plant metabolomics, metabolite identification, data visualisation, omics data, bioinformatics tools

## Abstract

The advancement of mass spectrometry technologies has revolutionised plant metabolomics research by enabling the acquisition of raw metabolomics data. However, the identification, analysis, and visualisation of these data require specialised tools. Existing solutions lack a dedicated plant-specific metabolite database and pose usability challenges. To address these limitations, we developed PlantMetSuite, a web-based tool for comprehensive metabolomics analysis and visualisation. PlantMetSuite encompasses interactive bioinformatics tools and databases specifically tailored to plant metabolomics data, facilitating upstream-to-downstream analysis in metabolomics and supporting integrative multi-omics investigations. PlantMetSuite can be accessed directly through a user’s browser without the need for installation or programming skills. The tool is freely available and will undergo regular updates and expansions to incorporate additional libraries and newly published metabolomics analysis methods. The tool’s significance lies in empowering researchers with an accessible and customisable platform for unlocking plant metabolomics insights.

## 1. Introduction

Metabonomics refers to the comprehensive analysis of endogenous metabolic small molecules in organisms, specifically studying the changes in metabolites within cells, tissues, or organs [1]. Plant metabolomics, a subset of metabonomics, focuses on identifying and quantifying metabolites involved in various cell regulatory processes, which comprise over 200,000 known plant metabolites, with each plant harbouring more than 5000 metabolites [2]. This field finds applications in studying plant responses to biotic and abiotic stresses, ensuring food safety, nutrition, pharmaceuticals, and cosmetics, as well as in supporting metabolic engineering and molecular breeding [3]. In addition, metabolomics is not only serving as a new technique in plant chemosystematics to play a role in the phylogenetic relationships of higher plants but is also contributing to the development of the new discipline of plant chemophenetics [4]. However, the study of plant metabolomics presents significant challenges due to their diverse chemical compositions and wide dynamic range. Several separation and detection technologies have been developed and applied to plant metabolomics research. Among them, liquid chromatography–mass spectrometry (LC–MS) has emerged as the dominant technique for metabolomics research due to its inherent advantages, including the absence of derivatisation requirements and the broader coverage of metabolite classes compared to gas chromatography [5]. LC–MS combines the separation power of LC with the high resolution and sensitivity of MS, enabling the detection and identification of a wide range of metabolites, including polar and non-polar compounds, small molecules, and larger biomolecules. This versatility has solidified the position of LC–MS as the preferred technique in plant metabolomics research, allowing for comprehensive metabolomic profiling and the study of complex biological systems.

While several methods and software tools have been developed for plant metabolomics research, no single metabolomic tool is applicable to the analysis of the entire metabolome due to the diverse nature of metabolites, significant variation in content across tissues and cells, and the complexity of the physical and chemical properties of the metabolites preventing the existence of a single tool applicable to the entire metabolome [6]. Thus, data obtained from metabolomics experiments require pre-processing using software packages such as XCMS [7], MS-DIAL [8] and MZmine2 [9]. Further cleaning of the metabolomics data, including missing value imputation, data standardisation, and quality assessment, can be performed using MetaboAnalystR [10]. Subsequently, metabolite identification, a critical step in plant metabolomics, is accomplished via comprehensive analysis combining the retention time (RT), mass-to-charge ratio and tandem mass spectrometry (MS/MS) matching [11]. XCMS is a web-based metabolomics analysis tool that allows users to upload raw data for data processing and statistical analysis, while also providing an R package for local implementation, enabling MS/MS-based metabolite peak identification and data statistics [7]. MS-DIAL, a user-friendly graphical interface-based metabolomics analysis software, supports targeted and untargeted LC–MS and GC–MS analysis. It not only comes with a wide collection of metabolomics and lipidomics spectral databases but also allows users to import their own libraries in MSP format for comprehensive searching and analysis of metabolites [8]. Following metabolite identification, univariate and multivariate analyses are conducted. Univariate analysis focuses on examining individual variables over a specific period. It involves the application of various statistical methods, such as the *t*-test, analysis of variance (ANOVA), and the Mann–Whitney U test. In contrast, multivariate analysis is more effective in exploring relationships between multiple variables and commonly employs principal component analysis (PCA), partial least squares discriminant analysis (PLSDA), and canonical correlation analysis (CCA). Notable open-source software packages, such as MetaboAnalystR [10] and MetaboDiff [12], facilitate both univariate and multivariate analyses in metabolomics studies.

Despite the success of combining various metabolomics analysis tools for plant metabolome analysis, the field still lacks a comprehensive and streamlined analytical platform that can effectively decode plant metabolites. Currently, such a platform is still in its developmental stage, necessitating the use of disparate software and databases [13]. To address this gap, we have developed PlantMetSuite, an online analytical platform that provides comprehensive compound annotation by integrating information from major databases. PlantMetSuite supports the analysis of plant metabolomics data, facilitates the accurate identification of plant metabolites and visualisation of data results, and simplifies routine analysis of plant metabolomics without the need for coding expertise. This platform is now accessible at https://plantmetsuite.verygenome.com/ (accessed on 24 July 2023).

## 2. Results

### 2.1. Workflow for the Metabolomics Analysis in PlantMetSuite

PlantMetSuite offers a comprehensive suite of visually oriented applications specifically designed for metabolomics analysis (Figure 1a), providing a wide array of functionalities. Each application in PlantMetSuite is accompanied by test data, sample results, and video tutorials to ensure user-friendliness and accessibility. These resources are designed to facilitate ease of use for all users. A detailed overview of the capabilities of each analysis application is presented in Appendix A. PlantMetSuite adopts a streamlined data-processing workflow comprising four primary steps (Figure 1b). To begin using PlantMetSuite, users are initially prompted to upload data in several formats. (1) Major MS vendor formats, including those from AB Sciex (.Wiff), Thermo Fisher Scientific (.RAW), Bruker Daltonics (.D), Agilent Technologies (.D), and Waters (.RAW). These data formats necessitate conversion to the mzXML and mgf formats using the msconvert application within PlantMetSuite or the ProteoWizard (version 3.0.22167) [14] installed locally by the user. (2) Pre-processed data files, including MS1 peak table files (in .CSV format or .XLS format) and MS2 spectrogram files (in .MSP format or .MGF format). MS1 files can be obtained from the xcms application in PlantMetSuite, XCMS Online [15], the locally installed XCMS by the user [7], MS-DIAL [8], or MZmine2 [9], while MS2 files can be derived from raw data conversion or MS-DIAL analysis. (3) Metabolite content tables suitable for analysis with various analytical software, including the MetAnno application within PlantMetSuite, XCMS Online [8], and MS-DIAL [8]. (4) Metabolite ID in the databases (KEGG, Metlin, PubChem, etc.) or the metabolite name. Furthermore, users are required to select the preferred database for metabolite identification. PlantMetSuite encompasses three database types. (1) The internally constructed standards database, incorporating *m*/*z*, MS/MS, and RT information derived from standards. (2) Public databases such as MoNA (http://mona.fiehnlab.ucdavis.edu), MassBank [16], HMDB [17], KEGG [18], Respect [19], and CASMI [20]. (3) The LC–MS-based MS/MS spectral tag database, encompassing identified and unidentified metabolic profiles of *Arabidopsis thaliana* cauline leaves, rosette leaves, stems, and inflorescence parts. This database proves valuable for investigating tissue specificity and novel functions of secondary metabolites in plants [21]. Subsequently, users selectively employ diverse applications as per their requirements for data analysis, including data pre-processing, data cleaning, metabolite identification, differential metabolite analysis, pathway analysis, and data visualisation. Finally, upon completion of the analysis, tabular results and vector graphics can be exported. Additionally, if provided, the results are conveyed to the user’s email address. The raw data utilised for the analysis are securely retained on the server for 24 h only, after which they are entirely purged to ensure data security.

### 2.2. Plant Metabolite Annotation Based on a High-Quality Plant-Specific Library

Metabolite annotation in metabolomics heavily relies on robust scoring of spectral matches between MS/MS spectra and library spectra. The fragmentation patterns observed during MS/MS are greatly influenced by the instrument platform and impact energy parameters utilized. Therefore, the construction of a reliable in-house standards library becomes indispensable to enhancing the precision of metabolite annotation. A detailed methodology outlining the curation of our high-quality plant-specific MS/MS spectral library is provided in the Methods section. Our in-house MS/MS spectral library includes a total of 1122 plant-specific metabolites, thus offering comprehensive coverage of plant-derived metabolites beyond what is available in public spectral libraries. To exemplify the identification process, we presented a detailed explanation using the identification of 1-O-beta-D-glucopyranosyl sinapate as an example (Figure 2a). In the negative ionisation mode, peak M386T77 exhibited a mass-to-charge ratio (*m*/*z*) of 385.1118 and an RT of 1.27 min. Employing a 25 ppm cutoff threshold, the MS1 identification confidently assigned this peak to 1-O-beta-D-glucopyranosyl sinapate. Fragment alignment between the experimental and standard spectra was performed using a 10 ppm threshold, where the aligned fragments were visually represented as dots, with their relative abundance distinguished by various colours. Subsequent scoring based on Equation (1) yielded an MS/MS score of 0.9606. The RT was assessed using the RT evaluation formula (Equation (2)), resulting in an RT score of 1. Integrating the scores using the score integration formula (Equation (3)), along with user-defined weights, yielded a final score of 0.9688. Consequently, the peak was conclusively identified as 1-O-beta-D-glucopyranosyl sinapate. The standards library incorporated into PlantMetSuite covers a wide range of plant metabolites, including carboxylic acids and derivatives (19%), organooxygen compounds (14%), fatty acyls (8%), flavonoids (4%), benzene and substituted derivatives (3%), organonitrogen compounds (3%), keto acids and derivatives (2%), and phenols (2%) (Figure 2b). Leveraging both positive and negative modes, our platform allows for the simultaneous quantification of numerous metabolites, reaching hundreds in a single analysis. Furthermore, representative metabolites from various classes are visually highlighted on the graph, with the colour-coded dots indicating their relative abundance (Figure 2c).

### 2.3. Metabolomics Analysis of Wild-Type Arabidopsis Thaliana and Ceramide Kinase Mutant acd5

Metabolomic profiling was conducted on wild-type *Arabidopsis thaliana* and the ceramide kinase mutant *acd5* using the aforementioned workflow. A total of 4831 peaks were detected in the positive mode, while 5459 peaks were detected in the negative mode. Among these peaks, 518 metabolites were identified, with 313 metabolites in the positive mode and 205 in the negative mode (Appendix A). PCA was employed, and the resulting scatter plot demonstrated the well-grouped biological replicates, indicating high experimental stability. Notably, a clear separation between the *acd5* samples and wild-type (Col-0) samples was observed in the plots, suggesting pronounced metabolomic differences between the mutant and wild-type plants (Appendix A). These findings were supported by the results of the partial least squares–discriminant analysis (PLS–DA) (Appendix A). Univariate analysis revealed 35 differentially abundant metabolites out of the 518 identified metabolites, with statistical thresholds set at a *p*-value < 0.05 and an absolute log2foldchange > 1 (Appendix A). The volcano plot illustrated the distribution of these significantly different metabolites (Figure 3b). To identify and characterise the dysregulated pathways, pathway enrichment analysis was performed. For each pathway, the hypergeometric test was used to calculate the corresponding *p*-value, with pathways exhibiting *p*-values below 0.05 (default threshold) considered enriched. A total of 10 dysregulated pathways were detected (Figure 3c), primarily associated with amino acid and sugar metabolism. Furthermore, several key differential metabolites underwent further re-examination (Figure 3d).

## 3. Discussion

Using PlantMetSuite for the analysis of untargeted LC–MS metabolomics data from wild-type *Arabidopsis thaliana* (Col-0) and the *acd5* mutant at 28 d after growth, we elucidated significant differences in the metabolic profiles of the *acd5* mutants, exhibiting a spontaneous programmed cell death phenotype, in comparison to their wild-type counterparts. A total of 30 metabolites exhibiting differential regulation were identified, followed by a comprehensive enrichment analysis of these metabolites. Among the notable findings, acetoacetyl-CoA, chlorophyll b, D-glucose, and proline were considerably diminished in the *acd5* mutants, while glutathione, pyridoxal 5’-phosphate, stearidonic acid, and UDP-D-glucose exhibited significant elevation. These findings signify that an insufficiency of acetylacetyl-CoA disrupts fatty acid synthesis and the biosynthesis of secondary metabolites in plants. The decrement in chlorophyll b corresponds to the *acd5* phenotype, while the reduced proline level and elevated glutathione level provide evidence of *acd5*-induced oxidative stress. Moreover, it has been previously reported that glutathione is involved in the signalling cascade of programmed cell death [22]. Furthermore, numerous metabolic pathways linked to sugar and amino acid metabolism were found to be disrupted, such as amino sugar and nucleotide sugar metabolism, pyruvate metabolism, starch and sucrose metabolism, glycolysis metabolism, glycolysis/gluconeogenesis, and terpenoid backbone biosynthesis. These discoveries contribute to our understanding of the mechanism underlying programmed cell death during sphingolipid accumulation in *acd5* mutants, underscoring the interplay between sphingolipid accumulation, metabolic disorders, and programmed spontaneous cell death.

In conclusion, PlantMetSuite is a web-based tool facilitating the simultaneous identification of hundreds to thousands of metabolites in plant samples, thereby empowering researchers to visualise plant metabolomics data. In this investigation, the metabolite identification adhered to the rigorous standards outlined by the Metabolomics Standards Initiative (MSI), employing either a standards database for level-one identification or a public database or MS2T database for level-two identification [23,24]. PlantMetSuite surpasses the prescribed orthogonality parameters established by the MSI through its integration of the precise mass, RT, and MS/MS fragment comparison. Our approach demonstrated comparable accuracy in identification and quantification compared to conventional methodologies while presenting distinctive attributes. These attributes include a high-quality standards database supplemented by an expansive public database that will be expanded in forthcoming updates. Moreover, PlantMetSuite offers a comprehensive suite of analytical procedures, enhancing versatility in analysing diverse plant metabolism data, along with supplementary visualization tools. Consequently, PlantMetSuite emerges as an indispensable tool for the analysis of metabolomics data in plants, catering to researchers with varying levels of bioinformatics expertise. PlantMetSuite is available at https://plantmetsuite.verygenome.com/.

## 4. Materials and Methods

### 4.1. Construction of Plant Standards Spectral Library

The construction of the standards spectral library in PlantMetSuite was based on a modified method derived from the NIST [25]. Employing Sciex TripleTOF 5600 instruments, MS2 spectra were acquired using commercially available metabolite standards. To acquire the MS2 spectra for each metabolite, a flow injection method was employed, utilizing target product ion scans. To ensure a robust representation, at least 16 MS2 spectra were selected for each metabolite. Clusters of MS2 spectra exhibiting high similarity (DP > 0.8) were carefully aggregated to generate a consensus MS2 spectrum. This acquisition process involved employing varying collision energies, including 10, 20, 30, 40, 50, 60, 70, and 35 ± 15 eV. The MS1 and RT data were obtained using the same LC–MS method described below. The current iteration of the plant standards spectral library in PlantMetSuite comprises a collection of 1122 metabolites, consisting of 1122 for the positive mode and 1119 for the negative mode.

### 4.2. Annotation Algorithms for Metabolites

Metabolite annotation in PlantMetSuite follows a systematic, algorithmic approach. Initially, the MS2 spectra were matched to the corresponding MS1 peaks based on their RT (±8 s) and *m*/*z* (±20 ppm) using the imported MS2 data and MS1 peak table. In cases where multiple MS2 spectra corresponded to a single MS1 peak, the algorithm selected the most abundant MS2 spectrum. To quantify the abundance of the MS2 spectrum, PlantMetSuite calculated the cumulative intensity of the top ten fragment ions, ensuring an accurate representation. When using MS-DIAL for peak detection and MS2 data output in the .MSP format, PlantMetSuite bypassed the MS2–MS1 matching step, thereby expediting the process. The resulting MS1/MS2 pairs were then compared to our in-house standard spectral library. This facilitated metabolite annotation. By default, a tolerance of ±25 ppm was employed for the MS1 *m*/*z* value. A tailored DP function was applied to assess the similarity between the experimental and standards spectra, yielding a similarity score ranging from 0 to 1 (Equation (1)). This score provides a quantitative measure of the degree of alignment, ranging from none to perfect alignment. For consistency, the fragment ion intensities in the MS2 spectra were rescaled, normalising the highest fragment ion to 1.
(1)$$\mathrm{score}.\mathrm{MSMS}=\frac{\sum \left(\left[\mathrm{intensity}.\mathrm{library}\right]\left[\mathrm{mz}.\mathrm{library}])([\mathrm{intensity}.\mathrm{experiment}\right]\left[\mathrm{mz}.\mathrm{experiment}\right]\right)}{\sqrt{\sum {\left(\left[\mathrm{intensity}.\mathrm{library}\right]\left[\mathrm{mz}.\mathrm{library}\right]\right)}^{2}{\left(\left[\mathrm{intensity}.\mathrm{experiment}\right]\left[\mathrm{mz}.\mathrm{experiment}\right]\right)}^{2}}}$$
where score.MSMS represents the score of MS2, and intensity.library and mz.library correspond to the abundance and the *m*/*z* of the product ions in the database, respectively. Similarly, intensity.experiment and mz.experiment represent the abundance and *m*/*z* of the product ions in the experimental data. Moreover, score.MSMS is derived from the combination of forward and reverse matches.

A trapezoidal function was used to evaluate the RT match score in PlantMetSuite (Equation (2)). This particular function has gained considerable acceptance in assessing the RT in gas chromatography analysis [26].
(2)$$\mathrm{score}.\mathrm{RT}=\left\{\begin{array}{l} 1,\left|\mathrm{RT}.\mathrm{experiment}\;-\;\mathrm{RT}.\mathrm{library}\right|\;<\;\mathrm{RT}.\mathrm{Tol}.\min\\ 1-\frac{\left|\mathrm{RT}.\mathrm{experiment}\;-\;\mathrm{RT}.\mathrm{library}\right|\;-\;\mathrm{RT}.\mathrm{Tol}.\min}{\mathrm{RT}.\mathrm{Tol}.\max\;-\;\mathrm{RT}.\mathrm{Tol}.\min},\mathrm{RT}.\mathrm{Tol}.\min\;\leq \;\left|\mathrm{RT}.\mathrm{experiment}\;-\;\mathrm{RT}.\mathrm{library}\right|\;\leq \;\mathrm{RT}.\mathrm{Tol}.\max\\ 0,\left|\mathrm{RT}.\mathrm{experiment}\;-\;\mathrm{RT}.\mathrm{library}\right|\;>\;\mathrm{RT}.\mathrm{Tol}.\max \end{array}\right\}$$
where score.RT represents the RT score, RT.library represents the RT in the database, RT.experiment represents the RT in the experiment, RT.Tol.min represents the minimum RT tolerance, and RT.Tol.max represents the maximum RT tolerance.

An integrated score, calculated by combining the scores from different dimensions, was determined using a linear weighting function, as described by Equation (3):score.integrate = Weight.MSMS × score.MSMS + Weight.RT × score.RT(3)
where score.RT and score.MSMS are the scores for the RT and MS/MS matches calculated in the previous steps. Weight.RT and Weight.MSMS are the weights assigned by the user for the RT and MS/MS match scores. Only metabolite candidates with integrated scores higher than the user-defined threshold were considered valid match results.

### 4.3. Software Implementation of PlantMetSuite

The interface of PlantMetSuite was implemented using Shiny Server (version 1.5.20.1002). The software tools in PlantMetSuite were developed using a combination of programming languages, including R, Perl, and Python. The web server hosting the PlantMetSuite platform is currently deployed on a high-performance Linux server boasting 32 cores (3.2 GHz CPU) and 256 GB RAM, ensuring optimal performance. Rigorous testing was conducted using the latest stable versions of web browsers, namely Chrome (114.0.5735.199), Firefox (115.0.2), and Microsoft Edge (44.19041.1.0).

### 4.4. Reagents, Plant Culture, and Sample Preparation

All the chemical standards were purchased from MedChemExpress (Monmouth Junction, NJ, USA), and a list of these standards can be found in Appendix A. Acetonitrile (ACN) and methanol (MeOH) of LC–MS grade were procured from Honeywell (Muskegon, MI, USA). Ammonium hydroxide (NH_4_OH) and ammonium acetate (NH_4_OAc) were obtained from Sigma-Aldrich (Burlington, MA, USA).

Wild-type *Arabidopsis thaliana* (Col-0) and *acd5* mutants [27] were stratified at 4 °C for 3 d and subsequently cultivated in a greenhouse under controlled conditions of 22 °C, 50% relative humidity, and a photoperiod of 16 h light and 8 h dark. The plants were illuminated with PAK lamps (PAK090311), providing a light intensity of 4800–6000 lux. Mature leaves of *Arabidopsis thaliana* were harvested and promptly placed in microcentrifuge tubes, which were immediately snap-frozen in liquid nitrogen and stored at −80 °C until further metabolite extraction. Prior to extraction, the plant samples were thawed on ice and homogenized using a Tissuelyser 192 (Jingxin Technologies, Shanghai, China) with beads (3 mm in diameter) and 200 μL of water (H_2_O) for homogenization. Subsequently, a mixture of ACN:MeOH (1:1, *v*/*v*; 800 μL) was added to the samples and vortexed for 30 s. The samples were then rapidly frozen for 1 min in liquid nitrogen, followed by thawing on ice. This vortexing, freezing, and thawing cycle was repeated three times. To precipitate proteins, the samples were incubated at −20 °C for 1 h, followed by centrifugation at 15,800× *g* for 15 min at 4 °C. The collected supernatant was then subjected to solvent evaporation using a vacuum concentrator (Labconco, Kansas City, MO, USA) until complete dryness was achieved. The resulting dried extracts were reconstituted by adding a mixture of ACN:H_2_O (1:1, *v*/*v*; 100 μL) and sonication (50 Hz, 4 °C) for 10 min. The solutions were centrifuged at 15,800 g for 5 min at 4 °C to remove any remaining insoluble debris. Prior to LC–MS/MS analysis, the clarified supernatants were transferred to glass HPLC vials for immediate LC–MS/MS analysis or temporary storage at −80 °C.

### 4.5. LC–MS/MS Analysis of Wild-Type Arabidopsis Thaliana and Ceramide Kinase Mutant acd5

Metabolomics data for the *Arabidopsis thaliana* samples were collected using a quadrupole time-of-flight mass spectrometer (TripleTOF 5600, AB SCIEX, Framingham, MA, USA) coupled with a UHPLC system (Shimadzu 30A; SHIMADZU Corporation, Kyoto, Japan). LC separation was achieved using a Waters ACQUITYUPLC BEHAmide column (particle size, 1.7 μm; 100 mm (length) × 2.1 mm (inner diameter)), with the column temperature maintained at 25 °C. Mobile phase A consisted of 25 mM ammonium hydroxide (NH_4_OH) + 25 mM ammonium acetate (NH_4_OAc) in water. Mobile phase B consisted of ACN, which was used for both the positive (ESI+) and negative (ESI-) modes. The gradient was configured as follows: with a flow rate of 0.3 mL/min, the elution conditions were set as follows: 0–1 min with 95% solvent B, 1–14 min with a linear decrease from 95% solvent B to 65% solvent B, 14–16 min with a linear decrease from 65% solvent B to 40% solvent B, 16–18 min with a constant 40% solvent B, 18–18.1 min with a linear increase from 40% solvent B to 95% solvent B, and finally, 18.1–23 min with a constant 95% solvent B. The injection volume was fixed at 2 μL, and all the samples were collected randomly for data acquisition. Information-dependent acquisition (IDA) was employed for the data collection. The source parameters were set as follows: gas 1 (GAS1) at 60 psi, gas 2 (GAS2) at 60 psi, curtain gas (CUR) at 30 psi, temperature (TEM) at 600 °C, declustering potential (DP) at either 60 V in positive mode or −60 V in negative mode, and ion spray voltage float (ISVF) at 5500 or −4000 V in positive or negative mode, respectively. The TOF MS scan parameters covered a mass range of 60–1200 Da, with an accumulation time of 200 ms and dynamic background subtraction enabled. For the product ion scans, the mass range was set to 25–1200 Da, with an accumulation time of 50 ms and a collision energy of 30 or −30 V in positive or negative mode. Additional parameters were the collision energy spread (0), resolution (units), charge state (1 to 1), intensity (100 cps), isotopic exclusion within 4 Da, mass tolerance (10 ppm), maximum number of candidate ions to monitor per cycle (6) and exclusion of previous target ions for 4 s after two occurrences.

### 4.6. Data Analysis

An MS1 peak table (.csv format) and MS2 data files (.mgf or .msp format) are required for import into PlantMetSuite. The MS1 peak table lists the metabolic peaks together with the annotated *m*/*z* and RT. This table was generated by extracting data from the raw MS files using popular peak-picking software like XCMS and MS-DIAL. The original MS files were converted into MS2 data files (.mgf format) using ProteoWizard (version 3.0.6150). Alternatively, users have the option to execute all the aforementioned steps within the PlantMetSuite framework. For data analysis using PlantMetSuite, users’ MS1 peak tables (.xls) and MS2 data (.mgf) are uploaded to PlantMetSuite for data analysis. Metabolite identification is performed based on user-defined parameters, encompassing the following specifications: polarity: positive/negative; instrument platform: Scienx; collision energy: 30; *m*/*z* tolerance: 20 ppm; RT match: checked; MS/MS match: checked; RT minimum tolerance: 30 s; RT maximum tolerance: 60 s; MS/MS mass range: 50–1250; MS/MS score cut-off: 0.6; RT score weight: 0.2; MS/MS score weight: 0.8; score cut-off: 0.6. Default parameters were used for all the other analyses, including data pre-processing, data cleaning, metabolite identification, differential metabolite analysis, pathway analysis, and data visualisation.

## Figures and Tables

**Figure 1 plants-12-02880-f001:**
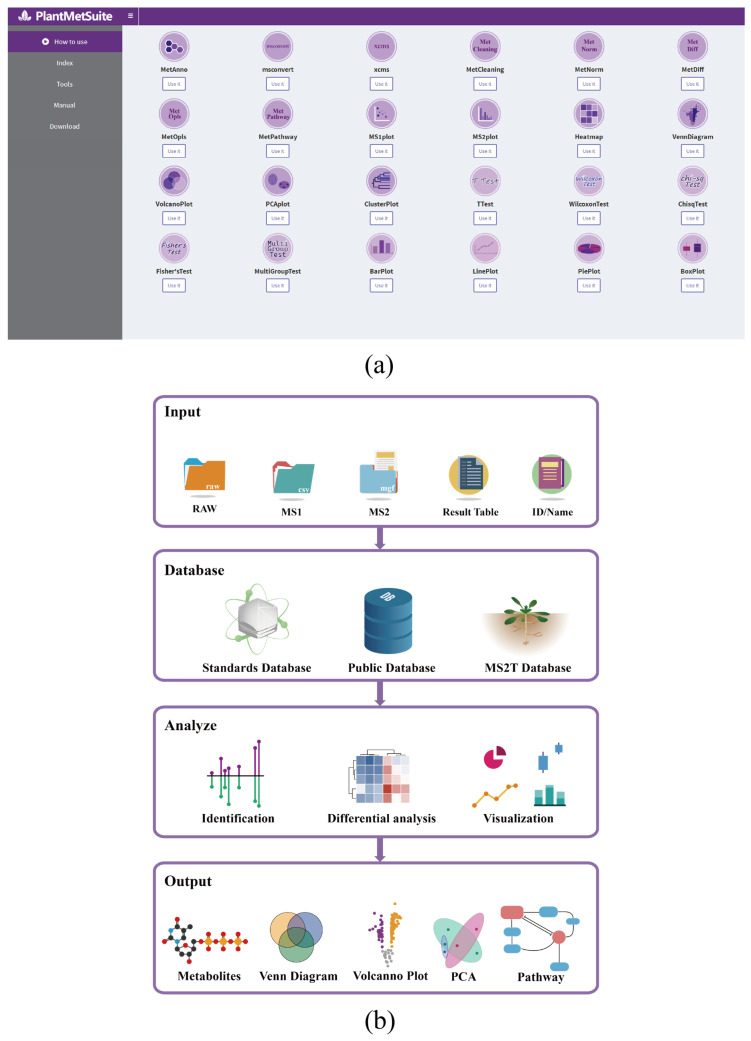
PlantMetSuite user interface design and functionalities. (**a**) Screenshot of the PlantMetSuite interface. Users can access an instructional video by clicking on the “How to use” button. The “Index” tab leads to the PlantMetSuite home page, the “Tools” tab provides direct access to the PlantMetSuite applications, the “Manual” tab allows users to consult the help manual, and the “Download” tab enables downloading of the data already analysed. (**b**) PlantMetSuite data-processing workflow. The analysis process consists of four steps: data entry, database selection, analysis tool selection, and result export. In the data entry step, users can input MS1 files, MS2 files, quantitative tables, and metabolite names or IDs. The database selection step allows users to choose from standards, public, and MS2T databases. The analysis tool selection step enables the selection of tools for metabolite identification analysis, difference analysis, and visualisation analysis. In the result export step, users can export various outputs, such as metabolite identification results, Venn diagrams, volcano diagrams, PCA results, and pathway enrichment results.

**Figure 2 plants-12-02880-f002:**
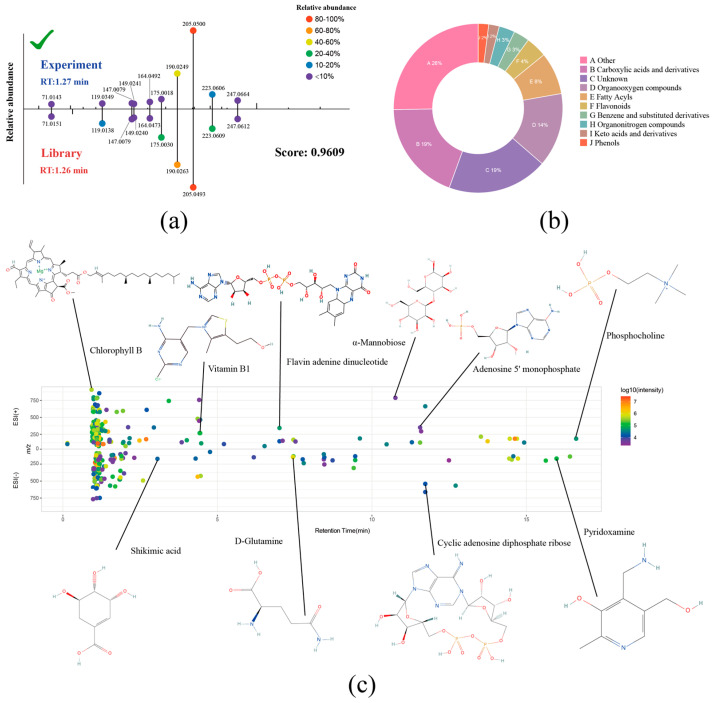
Metabolite identification process and capabilities of PlantMetSuite. (**a**) Illustration of the metabolite identification process using the example of 1-O-β-D-glucopyranosyl sinapate. The top part shows the spectrum from the experiment, and the bottom part displays the spectrum from the database. Colours represent the relative abundance of fragments, and the RT and the final scoring are marked on the image. (**b**) Pie charts depicting the mass proportions of different metabolite classes in the standard library. (**c**) PlantMetSuite enables the identification of hundreds of different plant metabolites in a single experiment.

**Figure 3 plants-12-02880-f003:**
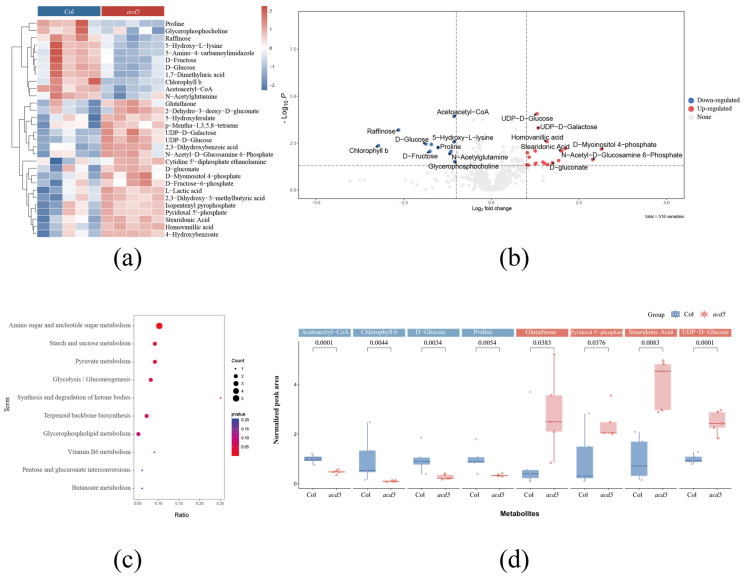
Differences in metabolomics between *acd5* and wild type. (**a**) Heatmap of the differentially regulated metabolites between wild type and *acd5*, as generated through hierarchical clustering. (**b**) A volcano plot illustrating the dysregulated metabolites in *acd5* (Student’s *t*-test, FDR-corrected *p*-values < 0.01). (**c**) Pathway enrichment analysis. The dot size corresponds to the number of metabolites in the pathway. (**d**) Representative metabolites that are up- and down-regulated in *acd5*, with different dots representing different biological repeats.

## Data Availability

All the pertinent data can be located in the article or Appendix A, and they are also accessible through the PlantMetSuite website (https://plantmetsuite.verygenome.com/ accessed on 24 July 2023). The PlantMetSuite source code is available on GitHub at https://github.com/YaonanLab/PlantMetSuite (accessed on 24 July 2023) for scientific research purposes. All other data in support of the results of this study are available at reasonable request from the corresponding author.

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
