# Peer review of "PlantMetSuite: A User-Friendly Web-Based Tool for Metabolomics Analysis and Visualisation"

_plants, 2023, doi:10.3390/plants12152880_

Round 1
Reviewer 1 Report
In my opinion:
I think that the idea of such a site is very good. I checked this website and it still has a lot of bugs. in my opinion, the scientific article should be published in another statistical journal.
Author Response
Thank you for pointing this out. In this version, we have fixed the bugs we have found so far, and in communicating with users, we have found that some bugs are caused by the uploaded data is not consistent with our requirements, we have recorded a video tutorial (https://plantmetsuite.verygenome.com/how_to_use.mp4) and made a manual (https://plantmetsuite.verygenome.com/manual.pdf), we hope it can answer your concerns. In addition, we thank for your suggestion about statistical journal.
Reviewer 2 Report
The manuscript describes a new web-based tool for metabolomics for plant metabolites. The manuscript is well written with only minor grammatical/spelling errors. The manuscript will be of interest to those in the plant metabolome research area, as there are minimal databases containing plant metabolites. The manuscript is suitable for publication following a few minor corrections.
Page 5 line 173. 'Metabolomic'
Page 6 lines 183, 188, and throughout the manuscript. 'p-value'
Page 7 line 249. ' method described'
Page 8. Insert increased line spacing after the formulas 1 & 2.
Page 9, lines 321 - 323. It is unclear if the clarified supernatants were stored at -80C until analysis or this was a step in the process prior to the analysis. If it is the later, how long were they placed at -80C before analysis. If it is for storage prior to analysis please adjust for clarity.
Page 9, line 330. 'NH4OH'
Page 9, line 331. 'NH4OAc'
Supplemental files could not be opened to be reviewed.
References are not in the proper format. Please format according to the Guide for Authors.
As a note to the authors regarding PlantMetSuite. To ensure the widest audience for use the manuals associated with the program should be supplied in multiple languages. I would suggest English be one of those as it is recognized as one of the scientific languages. Researchers will not want to cut and paste text from the program into translators to be able to resolve any questions that arise.
Please refer to comments and suggestion section regarding English corrections.
Author Response
The manuscript describes a new web-based tool for metabolomics for plant metabolites. The manuscript is well written with only minor grammatical/spelling errors. The manuscript will be of interest to those in the plant metabolome research area, as there are minimal databases containing plant metabolites. The manuscript is suitable for publication following a few minor corrections.
Page 5 line 173. 'Metabolomic'
[Response]:Thank you for pointing this out. We have replaced "metabonomic" with "metabolomic" in the new version (line 176, page 5).
Page 6 lines 183, 188, and throughout the manuscript. 'p-value'
[Response]:We are sorry for the errors. In the new version, we have replaced "P-value" with "p-value" throughout the manuscript (lines 187, 191, page 6, etc.).
Page 7 line 249. ' method described'
[Response]:We have added a space between "method" and "described" (line 252, page 7).
Page 8. Insert increased line spacing after the formulas 1 & 2.
[Response]:Thanks for pointing this out. We have added it.
Page 9, lines 321 - 323. It is unclear if the clarified supernatants were stored at -80C until analysis or this was a step in the process prior to the analysis. If it is the later, how long were they placed at -80C before analysis. If it is for storage prior to analysis please adjust for clarity.
[Response]:Thanks for pointing this out. We have changed it and it seems to be more accurate as follows (line 327-329, page 9): Prior to LC-MS/MS analysis, the clarified supernatants were transferred to glass HPLC vials for immediate LC-MS/MS analysis or temporary storage at - 80 °C.
Page 9, line 330. 'NH4OH'
[Response]:Thank you for your careful reading. We have corrected it (line 336, page 9).
Page 9, line 331. 'NH4OAc'
[Response]:Thank you for your careful reading. We have corrected it (line 337, page 9).
Supplemental files could not be opened to be reviewed.
[Response]:Thank you for pointing this out. We have checked the file and changed the compression format to a more common one (zip).
References are not in the proper format. Please format according to the Guide for Authors.
[Response]:Thank you for the careful reading. We have formatted it according to the Guide for Authors.
As a note to the authors regarding PlantMetSuite. To ensure the widest audience for use the manuals associated with the program should be supplied in multiple languages. I would suggest English be one of those as it is recognized as one of the scientific languages. Researchers will not want to cut and paste text from the program into translators to be able to resolve any questions that arise.
[Response]:Thank you for the valuable comments and suggestions. In this new version we have set the default language of the website to English. The manual and video tutorial are also available in English now:
Video tutorial: https://plantmetsuite.verygenome.com/how_to_use.mp4
Reviewer 3 Report
The development of plant metabolomics is very relevant today. The data obtained as a result of the analysis of a set of metabolites in plants have both fundamental and practical meaning. The article is timely and contains enough new information. The analysis was carried out at the highest level. The authors investigated the metabolomes of poorly studied plants Arabidopsis thaliana and the ceramide kinase mutant acd5 and invite all interested people to use the database and site on plant metabolomics "PlantMetSuite" created by them. In fact, this is important and relevant today - the development of not only plant genomics and transcriptomics, but also their metabolomics. Many thanks to the authors for an interesting article! Some general remarks and suggestions for the work and the article:
- The practical output of plant metabolomics is displayed in the article well (Lines 32-40). Please add a couple of sentences about the fundamental side of these studies. Plant metabolomes are used when comparing closely related species, this opens up a new round of development for the currently slightly forgotten plant chemosystematics and the use of metabolimics data in the study of plant phylogeny and evolution.
- The site is not all in English. We really want to use this site, but there is a language barrier.
- Less common compounds, such as chromones and furochromones, are not in the database of substances (Table S2). Surely this will be done in the future.
Author Response
The development of plant metabolomics is very relevant today. The data obtained as a result of the analysis of a set of metabolites in plants have both fundamental and practical meaning. The article is timely and contains enough new information. The analysis was carried out at the highest level. The authors investigated the metabolomes of poorly studied plants Arabidopsis thaliana and the ceramide kinase mutant acd5 and invite all interested people to use the database and site on plant metabolomics "PlantMetSuite" created by them. In fact, this is important and relevant today - the development of not only plant genomics and transcriptomics, but also their metabolomics. Many thanks to the authors for an interesting article! Some general remarks and suggestions for the work and the article:
- The practical output of plant metabolomics is displayed in the article well (Lines 32-40). Please add a couple of sentences about the fundamental side of these studies. Plant metabolomes are used when comparing closely related species, this opens up a new round of development for the currently slightly forgotten plant chemosystematics and the use of metabolimics data in the study of plant phylogeny and evolution.
[Response]:Thank you for your valuable comments and suggestions. We have added a sentence in the introduction sections of the new version as follows: In addition, metabolomics is not only serving as a new technique in plant chemosystematics to play a role in the phylogenetic relationships of higher plants, but is also contributing to the development of the new discipline of plant chemophenetics.
- The site is not all in English. We really want to use this site, but there is a language barrier.
[Response]:Thank you for your valuable comments and suggestions. We have changed the default website language to English in this new release. The manual and video tutorial are now available in English :
Video tutorial:https://plantmetsuite.verygenome.com/how_to_use.mp4
- Less common compounds, such as chromones and furochromones, are not in the database of substances (Table S2). Surely this will be done in the future.
[Response]:Thank you for your suggestion. Chromones and furochromones are important secondary metabolites in plants with diverse roles in defence, stress tolerance, reproduction and interactions with the environment. We will add these compounds to our database in the next update of PlantMetSuite.